# Image as a World: Generating Interactive World from Single Image via Panoramic Video Generation

Dongnan Gui[1]*    Xun Guo[2]    Wengang Zhou[1]    Yan Lu[2]

[1]University of Science and Technology of China    [2]Microsoft Research Asia

{gdn2001@mail., zhwg@}ustc.edu.cn {xunguo, yanlu}@microsoft.com

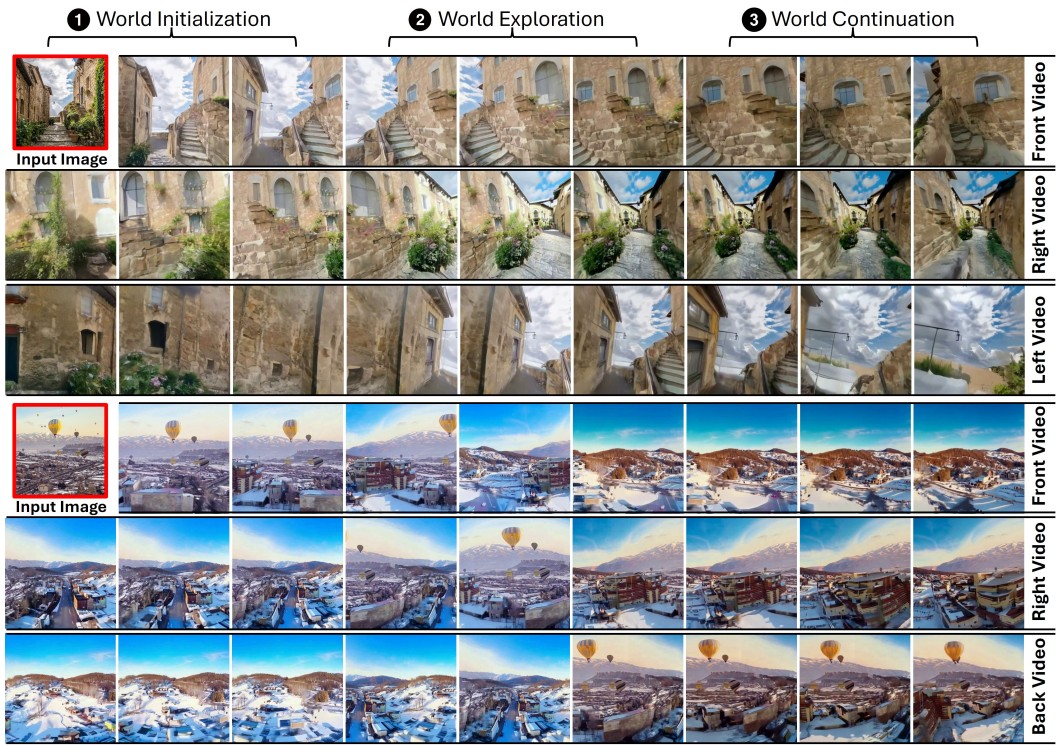

Figure 1: Generated world by our IaaW. Each line represents a fixed-view video that includes our proposed three stage of visual world generation: initialization, exploration, and continuation.

## Abstract

Generating an interactive visual world from a single image is both challenging and practically valuable, as single-view inputs are easy to acquire and align well with prompt-driven applications such as gaming and virtual reality. This paper introduces a novel unified framework, Image as a World (**IaaW**), which synthesizes high-quality 360-degree videos from a single image that are both controllable and temporally continuable. Our framework consists of three stages: world initialization, which jointly synthesizes spatially complete and temporally dynamic scenes from a single view; world exploration, which supports user-specified viewpoint rotation; and world continuation, which extends the generated scene forward in

*Work done during internship at Microsoft Research Asia.

39th Conference on Neural Information Processing Systems (NeurIPS 2025).

time with temporal consistency. To support this pipeline, we design a visual world model based on generative diffusion models modulated with spherical 3D positional encoding and multi-view composition to represent geometry and view semantics. Additionally, a vision-language model (IaaW-VLM) is fine-tuned to produce both global and view-specific prompts, improving semantic alignment and controllability. Extensive experiments demonstrate that our method produces panoramic videos with superior visual quality, minimal distortion and seamless continuation in both qualitative and quantitative evaluations. To the best of our knowledge, this is the first work to generate a controllable, consistent, and temporally expandable 360-degree world from a single image.

# 1   Introduction

Recent advances in world models [11, 21] and video generation have enabled simulation and extension of environments in rich, multimodal ways. World models have evolved to handle raw visual inputs, producing videos conditioned on actions or inferred intent across domains such as robotics [41, 18], autonomous driving [48, 10], and interactive gaming [4]. Concurrently, large-scale diffusion models [17, 7, 28, 43] and vision transformers [27] have redefined the frontier of video generation, achieving high fidelity and temporal coherence across diverse conditions. These advancements collectively point toward a promising new direction: building dynamic, controllable, and immersive environments directly from visual cues.

In this work, we take a step further by proposing a novel problem: generating an explodable and temporally extendable panoramic world from a single image—one that not only predicts future frames, but also supports interactive viewpoint control, enabling arbitrary view rotations and continuous scene evolution. Compared to prior methods that rely on multi-view or panoramic input, our single-image setup significantly reduces the cost of data acquisition and aligns with the growing trend of prompt-based generative models. Unlike traditional world models that focus on action-conditioned prediction, our approach synthesizes immersive scenes that respond to user-specified actions, enabling both free-form viewpoint control and continuous scene expansion.

Generating such a visual world from single image is both practically appealing and technically challenging. It requires the model to infer latent geometry, spatial layout, and temporally coherent dynamics from highly limited visual evidence—an under-constrained and ill-posed task. We formulate this as a new direction in panoramic video generation, where the goal is to synthesize panoramic, navigable, and temporally extensible video from minimal visual input.

Existing methods are not designed for this setting. Many prior approaches to panoramic video generation adopt one-shot generation strategies without temporal continuity or interaction capability. For instance, 360DVD [38] relies on text-to-video models with limited resolution, while 4K4DGen [22] generates each frame independently without temporal coherence. Others assume richer input such as multi-view videos [42, 25] or full panoramic images [20, 23].

To address these challenges, we structure our solution as a three-stage generative pipeline shown in Fig. 2: (1) World Initialization, which synthesizes a spatially complete and temporally coherent panoramic video from a single image, which provides stable spatiotemporal foundation for the subsequent stages; (2) World Exploration, which enables interactive navigation by modeling viewpoint changes as actions, thereby embedding user control directly into the generation process; and (3) World Continuation, which extends the scene forward in time while maintaining temporal consistency beyond a fixed horizon. Each stage addresses a specific limitation in prior work, and they collectively enable consistent, controllable and infinitely extensible world synthesis. To support this pipeline, we design a *visual world model*, implemented by augmenting a diffusion-based video generator with 3D Spherical Rotary Positional Encoding (RoPE) and multi-view composition. These components equip the model with the ability to represent scene geometry and maintain diversity across dynamic panoramic sequences. To enhance controllability and prompt alignment, we also finetune a vision-language model (IaaW-VLM) that generates semantically grounded, view-specific prompts conditioned on the user's perspective. Comprehensive experiments demonstrate that IaaW is capable of generating high-fidelity, semantically plausible panoramic videos that are both spatially coherent and temporally smooth. To our knowledge, this is the first framework to achieve infinitely expandable, user-controllable panoramic world synthesis from a single image.

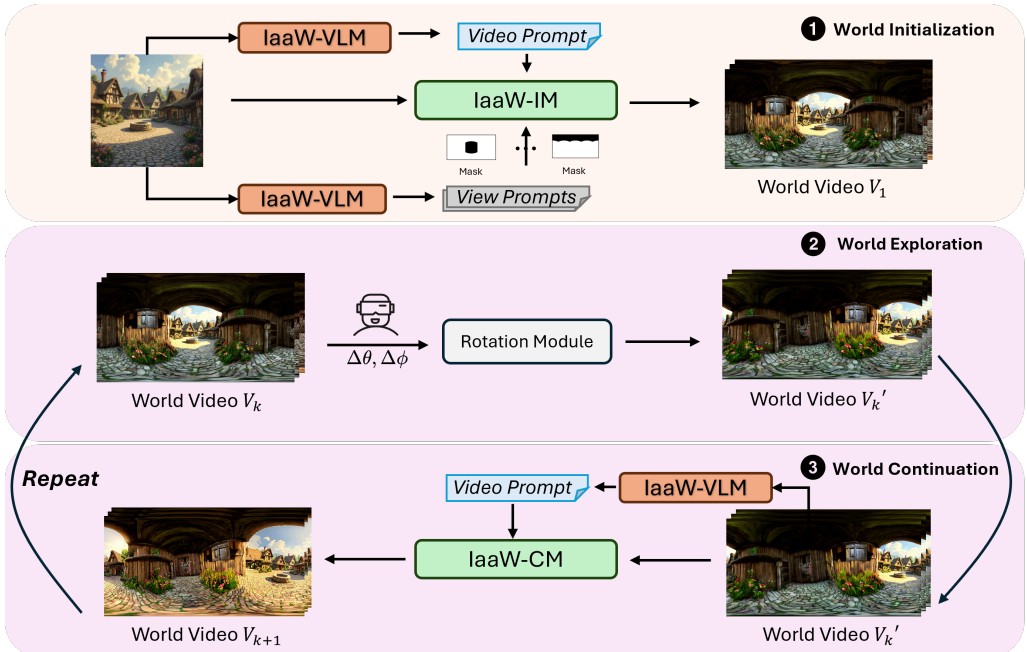

Figure 2: Pipeline of our proposed IaaW method, which consists of three core stages: world initialization, world exploration, and world continuation. In the world initialization stage, given a single reference image, we employ our finetuned IaaW-VLM to produce a holistic video-level prompt alongside multiple view-specific prompts. These, in conjunction with the input image, are processed by our IaaW-InitialModel (IaaW-IM) to generate the initial world video $V_1$. World exploration stage enables user's spatial control over the generated scene, the rotation module transforms the video $V_k$ to reflect the desired viewpoint video $V_k'$. In the final world continuation stage, the rotated video and its associated prompt are fed into the IaaW-ContinualModel (IaaW-CM), which produces an extended segment of the world. This process is inherently recursive, allowing the newly generated video to undergo further view rotations and extensions.

## 2 Related Work

### 2.1 World Model

World models [11, 21] aim to predict the future evolution of an environment in response to specific actions. Traditionally, these models operated in abstract spaces and were predominantly used for planning [14, 31, 30] or policy learning [13] in reinforcement learning contexts. Recent advances in generative modeling have extended world models to the visual domain, enabling video generation conditioned on control inputs [49]. In autonomous driving [48, 10], models predict based on driver actions, while in robotics [41, 18], predictions are conditioned on control signals of robots. Genie [4] further generalizes this by learning action-conditioned dynamics from raw gameplay videos in an unsupervised manner and WonderWorld [44] focuses on static 3D scene world generation using Gaussian-Splatting-like representation from a single image. Notably, 3D-based world generation like WonderWorld [44] does not produce equirectangular video but instead reconstructs a 3D world scene using Gaussian splatting or other representations, which may enable longer exploration paths but often suffers from artifacts inherent to splatting such as blur and point cloud sparsity. In contrast, our method generates temporally consistent and spatially coherent dynamic video with a spherical field of view, offering a more immersive and artifact-free experience. Leveraging the interactive nature of the panoramic videos, we treat user-specified view rotations as world actions and generate the corresponding next-step visual evolution, enabling immersive, controllable, and continuable world generation.

## 2.2 Video Generation

Recent advancements in diffusion models [33, 17, 34, 7, 28, 26] have propelled video generation, with hierarchical U-Net [29] and diffusion vision transformer (DiT) [27] architectures leading the way in spatiotemporal modeling. Approaches like Imagen Video [16] and Make-A-Video [32] extend these models with temporal attention, while Sora [3] scales diffusion transformers for video synthesis. In the open-source landscape, CogVideoX [43] introduces a 3D causal VAE with adaptive LayerNorm for efficient spatiotemporal modeling , and Hunyuan Video [39] employs a dual-stream transformer for enhanced text-video alignment . Wan [37] addresses high-resolution generation with dynamic 3D-VAE compression. These models highlight the growing trend toward hybrid architectures and scaling strategies for balancing fidelity and efficiency.

## 2.3 Panoramic Video Generation

Recent advances in panoramic video generation explore diverse paradigms [38, 25, 20]. 360DVD [38] is the first to tackle text-to-panoramic video synthesis by integrating a 360-adapter into early-stage T2V pipelines. However, it is limited by low-resolution training data and underpowered base models, yielding suboptimal quality. 4K4DGen [22] proposes a training-free approach for animating 4K panoramic images by independently rendering perspective views and spatially fusing them, while OmniDrag [23] enables interactive control via drag-based motion manipulation. DynamicScaler [20] enhances spatial scalability using an offset-shifting denoiser to synthesize spherical panoramas, followed by a learned upscaling stage. Several works address panoramic video generation by outpainting from nFOV inputs. VideoPanda [42] introduces multi-view attention to maintain spatiotemporal consistency, whereas [25] reframes the task as video-to-video generation. In driving applications, Panacea [40] leverages BEV representations for conditional panoramic synthesis. In contrast, our method directly generates high-fidelity, temporally coherent and continuable panoramic videos from a single image, achieving both spatial diversity and temporal infinity without auxiliary inputs.

# 3 Method

## 3.1 Visual World Model

Our visual world model is built on a diffusion-based video backbone, enhanced with multi-view composition and 3D Spherical RoPE. To support different goals, we introduce two finetuned variants: IaaW-InitialModel (IaaW-IM) for world initialization and IaaW-ContinualModel (IaaW-CM) for world continuation. While sharing the same architecture, the two models are optimized for different stages, which are scene reconstruction from a single image vs. temporally coherent extension.

### 3.1.1 Multi-View Composition

To address the limitations of one-shot conditioning in existing video generation models, we propose a multi-view composition method that significantly enhances the quality and diversity in world initialization. This mechanism takes two inputs, view masks, which are binary masks that correspond to user-specified predefined views (e.g., front, left, top), and view prompts, which are prompts aligned with each masked region generated by our IaaW-VLM, providing textual guidance for content generation from that specific viewpoint. As depicted in Fig. 3, our method begins with a single reference image, a

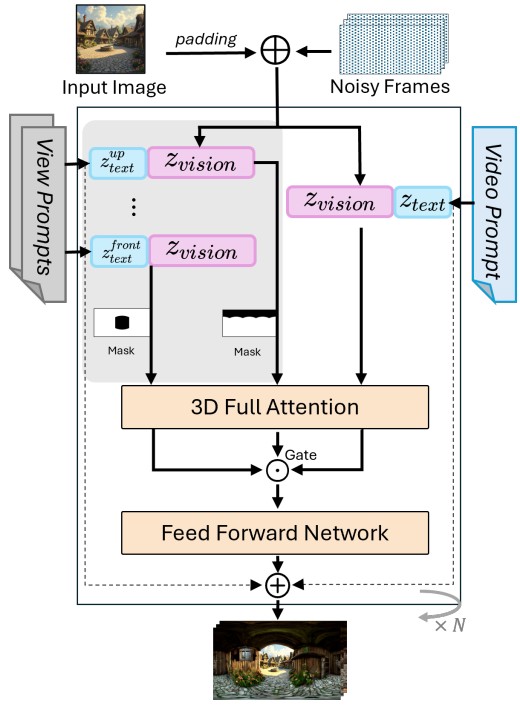

Figure 3: Multi-view composition used in IaaW-IM's MM-DiT blocks in world initialization.

corresponding video prompt, and a set of auxiliary view prompts, each of which is paired with spatially aligned masks. These view prompts are semantically and spatially diverse renderings of the scene, intended to provide additional geometric and contextual priors.

Building on recent powerful video generative models, such as CogVideoX [43], which uses MM-DiT blocks [8] and concatenate textual ($z_{\text{text}}$) and visual ($z_{\text{vision}}$) features, we introduce a multi-view conditioning mechanism for improved world initialization. Here, $z_{\text{text}}$ comes from the main video prompt, and $z_{\text{vision}}$ encodes a padded reference image with noisy frames. We add a parallel attention path using view-aware features: $z_{\text{text}}^{\text{view}}$ from IaaW-VLM is concatenated with $z_{\text{vision}}$ and modulated by view masks for localized 3D full attention. This stream runs in parallel with the base attention and is adaptively gated to fuse multi-view cues with global context. For clarity, AdaLN and scale-and-shift components are omitted from Fig. 3.

### 3.1.2 3D Spherical RoPE

We propose a unified 3D Spherical Rotary Positional Encoding (RoPE) that extends traditional rotary embeddings [35, 43] to spherical video domains. By embedding positional information in both spherical space [6, 45, 47] and time, our method aligns with the geometric structure of equirectangular panoramic video while preserving the rotation-equivariant properties of RoPE.

Let a video $V \in \mathbb{R}^{H \times W \times D \times T}$ represent a sequence of frames with height $H$, width $W$, feature dimension $D$, and temporal length $T$. Each spatial coordinate $(x, y)$ is mapped to spherical angles via:

$$\theta = \frac{\pi}{2}\left(\frac{2y}{H} - 1\right), \quad \phi = \pi\left(\frac{2x}{W} - 1\right), \tag{1}$$

where $\theta$ and $\phi$ denote latitude and longitude, respectively. We then construct a unified 3D positional encoding by modulating angular and temporal components in a factorized trigonometric basis:

$$\text{RoPE}_{x,y,t,d} = [\cos(2^d\theta) \cdot \cos(2^d\phi) \cdot \cos(2^d \cdot 2\pi t), \sin(2^d\theta) \cdot \cos(2^d\phi) \cdot \cos(2^d \cdot 2\pi t), \dots] \tag{2}$$

which compactly encodes the 3D positional across spatial angles $(\theta, \phi)$, frequency $d$ and normalized time $t$. 3D Spherical RoPE captures rotational symmetries on the spherical surface while enabling temporal phase alignment, resulting in a compact and geometry-aware encoding mechanism for panoramic video generation.

## 3.2 IaaW Pipeline

### 3.2.1 World Initialization

World Initialization serves as the entry point for visual world synthesis, which establishes the spatiotemporal foundation for subsequent user-controlled exploration and continuation. Given only a single-view image, the model must generate an initial panoramic video clip that is both spatially complete and semantically coherent, despite the severe ambiguity posed by missing multi-view context.

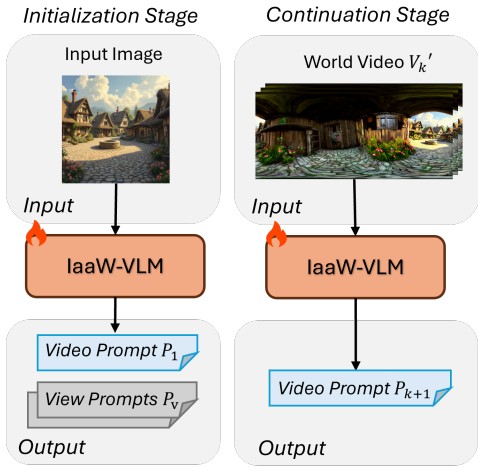

Figure 4: Functionality of IaaW-VLM.

To enhance semantic suitability and consistency in video generation, we introduce a world context model IaaW-VLM that generates both global and view-specific prompts. For each equirectangular video $V$, we first employ a caption model for a global prompt $P$ that summarizes the entire scene. The video is then spatially segmented into multiple views $\{V_v\}$ and individually captioned to yield prompts $\{P_v\}$, capturing the localized context. From each $V_v$, we extract a representative frame $I_v$, which forms the dataset $\{V, P, \{V_v, P_v\}, \{I_v\}\}$. This corpus supports training for IaaW-VLM, whose functionality is shown in Fig. 4. IaaW-VLM can generate $\{P_v\}, P$ from single view image $I_v$ and $P$ from video $V$,

which supports IaaW-IM and IaaW-CM separately. By grounding generation in this multi-granular context, IaaW-VLM acquires an enriched understanding of both spatial structure and temporal coherence.

With visual world model IaaW-IM and above IaaW-VLM, as shown in Fig. 2, we send entire video prompt and view prompts with corresponding masks into IaaW-IM model. This process generates the world video as

$$V_1 = \mathcal{IM}(I, P_1, \{M_v, P_v, v \in \text{views}\}) \tag{3}$$

where $P_1$ and $P_v$ represent initial prompt and view prompts respectively, $M_v$ represents masks.

### 3.2.2 World Exploration

Panoramic video enables immersive navigation by allowing users to rotate their virtual viewpoint within a spherical environment. We model this interaction as a transformation in spherical coordinates applied to the $k$th equirectangular video $V_k \in \mathbb{R}^{H \times W \times D \times T}$, where $W = 2H$, and $D, T$ denote the channel and temporal dimensions. Each pixel $(x, y) \in [0, W) \times [0, H)$ corresponds to spherical coordinates $(\theta, \phi)$ following Eq. (1). These angles represent latitude $\theta \in [-\frac{\pi}{2}, \frac{\pi}{2}]$ and longitude $\phi \in [-\pi, \pi)$. User-specified pitch and yaw rotations $(\Delta\theta, \Delta\phi) \in \mathbb{R}^2$ simulate view changes by adjusting the angles:

$$\theta' = \text{clip}(\theta + \Delta\theta, -\tfrac{\pi}{2}, \tfrac{\pi}{2}), \quad \phi' = \phi + \Delta\phi \tag{4}$$

Here, $\text{clip}$ ensures that the elevation stays within the bounds of the spherical domain. To map back to image coordinates, we have

$$x' = \left(\frac{\phi'}{2\pi} + \frac{1}{2}\right) W \mod W, \quad y' = \left(\frac{\theta'}{\pi} + \frac{1}{2}\right) H \tag{5}$$

The complete process yields the rotated video $V_k' \in \mathbb{R}^{H \times W \times D \times T}$, which is obtained by sampling the original video at $V_k(x', y', :, t)$ for each $(x, y, t)$.

### 3.2.3 World Continuation

View-aware world continuation stage enables the synthesis of temporally extended and visually coherent video sequences conditioned on a user-defined reference view. Our approach is built upon the visual world model IaaW-CM, which operates in an autoregressive manner, progressively generating video segments while maintaining view and content consistency over time in Eq. (6).

$$V_{k+1} = \mathcal{CM}(V_k', P_{k+1}) \qquad k = 2, \ldots, n \tag{6}$$

Specifically, following the paradigm of IaaW-IM, we substitute the single view image with video $V_k'$ from previously rotated video chunk. At step $k$, the IaaW-VLM produces the next prompt $P_{k+1}$ based on the evolving visual context, guiding the generation of segment $V_{k+1}$ towards arbitrary length. This stage establishes a foundation for open-ended and infinite scene generation, where a coherent and semantically meaningful world can emerge over extended temporal horizons, grounded in a user-defined viewpoint trajectory.

## 4 Experiments

### 4.1 Experimental Setup

**Models**  In the field of video generation, there are few open-source video diffusion models available for experimentation. We use CogVideoX1.5-5B-I2V [43], a text-image conditional video generator that supports arbitrary resolution and is well suited to our 2:1 aspect-ratio video setup. We use equirectangular videos to finetune IaaW-IM, where the input image is padded before being fed into the model. IaaW-CM is finetuned on top of IaaW-IM, using the previous video chunk as input and the next video chunk as output. Finetuning is conducted over two weeks on 4×A100 GPUs, followed by one week of progressive finetuning. Due to the absence of released code from prior panoramic methods [20, 25, 22], we implemented two baselines for comparison. One is 360I2V, a panoramic animation baseline fine-tuned from CogVideoX, which takes panoramic image as input to generate panoramic videos. Another is FETA (First Expand, Then Animate), a two-stage baseline for world

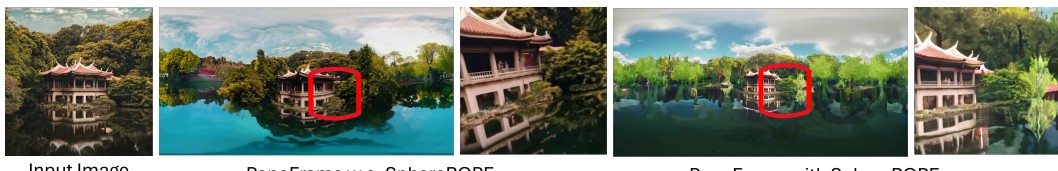

Input Image         PanoFrame w.o. SphereROPE          PanoFrame with SphereROPE

Figure 5: The results of reducing distortions of 3D Spherical RoPE in world initialization.

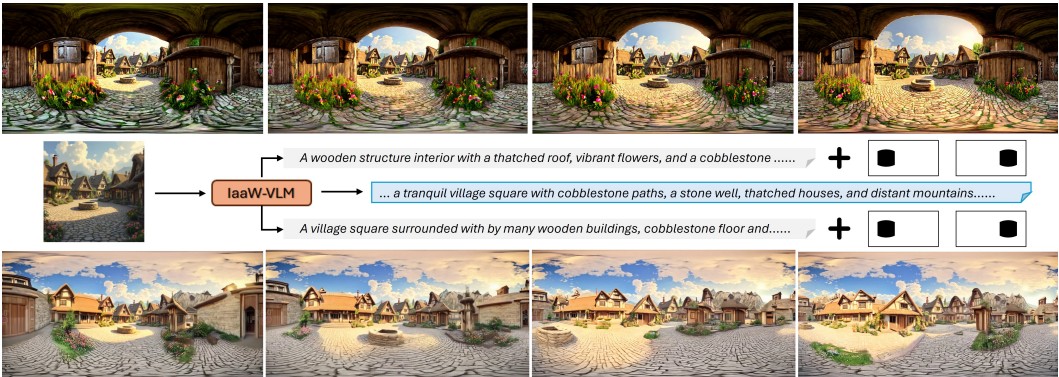

Figure 6: Generation results of multi-view composition for world initialization.

initialization, where we combine Diffusion360 [9] for NFoV-to-panorama expansion with 360I2V for subsequent animation. We compare our world initialization results with FETA, compare the world continuation results with 360I2V, and compare our whole IaaW pipeline with FETA+360I2V.

**Data**    We consider several panoramic video datasets, including WEB360 [38] and 360-1M from ODIN [36]. Due to WEB360's limited scale and low resolution (2K videos at $1024 \times 512$), it is excluded from our study. From 360-1M, we curate a high-resolution, equirectangular subset by filtering out static scenes and selecting diverse, dynamic content. Captions are generated using Qwen-VL-2.5 [2], and low-quality samples are removed based on caption quality. Augmented with an internal collection, our final dataset comprises 120K videos at $2048 \times 1024$ resolution. An 8K high-quality subset is further collected for progressive finetuning.

**Metrics**    To evaluate video generation quality, we consider both overall and per-view fidelity and consistency using metrics from VBench [19] and VideoBench [15]. *Subject Consistency* measures temporal coherence via the average cosine similarity of DINO [5] features between each frame and the first. *Motion Smoothness* is quantified by the mean absolute error between interpolated and dropped frames, while *Aesthetic Quality* is predicted using the LAION aesthetic model [1]. *Video-Text Consistency* assesses semantic alignment with the prompt, computed as the average score (1–5) assigned by a vision-language model. To evaluate continuous generation results, we concatenate videos from preceding steps, rotational transitions, and subsequent generations to evaluate coherence over extended sequences.

## 4.2 Qualitative Analysis

**World Initialization Results**    We first demonstrate the effectiveness of our 3D spherical RoPE in Fig. 5. The figure compares panoramic frame produced by our IaaW-IM. When rendering the panoramic image from a specified viewpoint, the model with spherical RoPE exhibits fewer distortions. In particular, it preserves the correct perspective geometry of structures such as the pavilion, whereas the model without that yields deformed objects with incorrect perspective relationships.

To assess the impact of multi-view composition in our initialization model IaaW-IM, we visualize generation results under varying view prompts in Fig. 6. Using a fixed video prompt and identical spatial masks, we observe that distinct view prompts (e.g., wooden buildings vs. flower yards) yield semantically diverse scene expansions. This demonstrates the fine-grained controllability afforded by

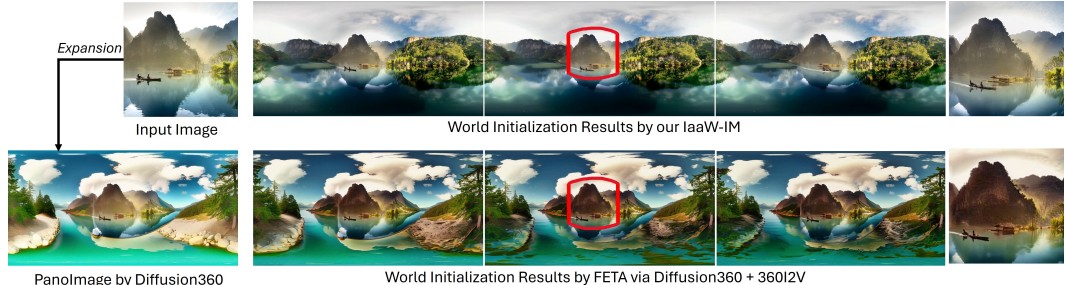

Figure 7: World initialization results compared with First Expanding Then Animating(FETA).

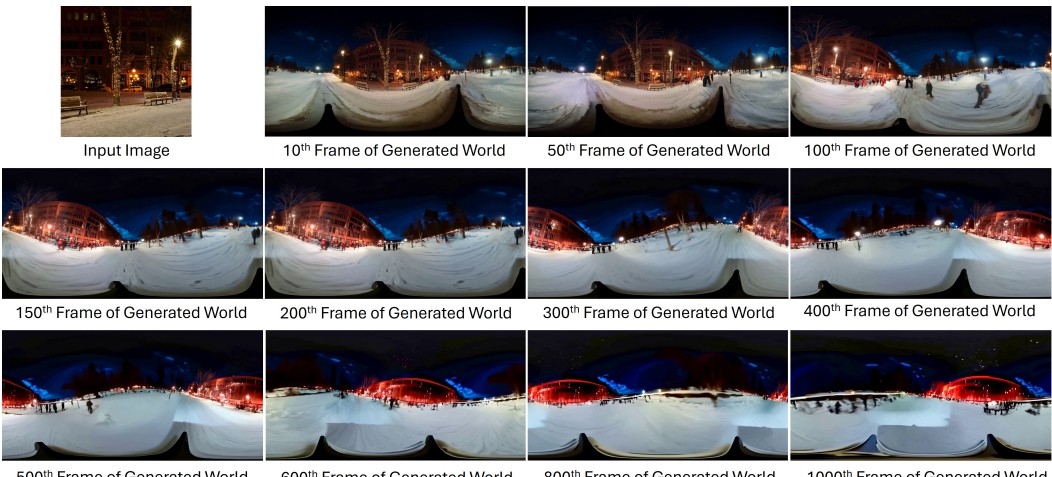

Figure 8: Long panoramic world video generated by our IaaW.

multi-view composition mechanisms and underscores their ability to guide content-specific scene synthesis during world initialization.

In Fig. 7, we compare our initialization strategy with a baseline that first performs panoramic extrapolation then animates the results in two separate stages. This decoupled spatial-temporal generation often leads to pronounced spatial artifacts and temporal discontinuities. In contrast, our method jointly models spatial structure and temporal dynamics and delivers coherent expansions that maintain global scene structure while enabling temporally smooth motion, establishing a superior world base.

We also include an example of a long panoramic world video generated by our IaaW in Figure Fig. 8. Our three-stage method successfully converts a single input image into a relatively long panoramic world. Specifically, our chunk-by-chunk method maintains high-quality results within the first minute or approximately ten rounds. The results begin to drift during super-long continuations like beyond minutes, leading to poor and vague content. Addressing this long-range coherence issue is a core problem across the field and is reserved for future work.

**World Continuation Results**   We evaluate the continuation model IaaW-CM in Fig. 9, where the initialized world is an aerial view towards a lighthouse. Our model maintains directional consistency across extended sequences after rotational transformations. Specifically, our generated continuation video persistently advances toward the lighthouse while remaining both temporally stable and spatially coherent. In comparison, the baseline, which conditioned solely on the last frame, suffers from abrupt motion discontinuities and visual degradation, and fails to preserve global motion dynamics. These findings highlight the efficacy of our IaaW-CM in capturing long-consistent motion trajectories.

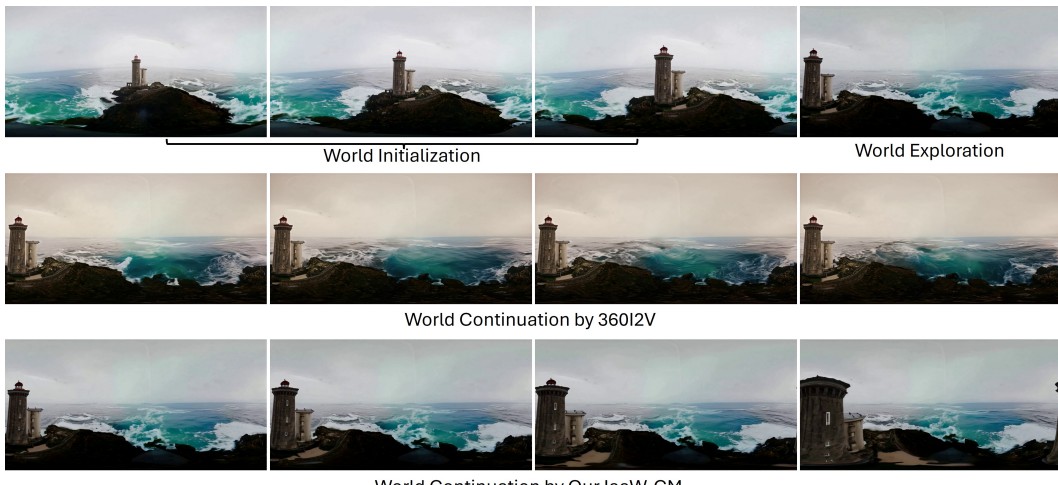

World Initialization        World Exploration

World Continuation by 360I2V

World Continuation by Our IaaW-CM

Figure 9: World continuation results of our IaaW-CM compared with baseline 360I2V.

Overall, our IaaW framework demonstrates qualitatively superior results in both initialization and continuation stages, affirming its effectiveness in generating visually coherent, controllable, and temporally consistent panoramic video worlds.

## 4.3 Quantitative Analysis

We present the quantitative results in Table 1, where we evaluate the videos in three setups: world initialization videos, world continuation videos, and entire world videos. The latter refers to the concatenated video of initialization video, world exploration video, and world continuation video. We evaluate the results using two distinct methods: "All", which assesses the entire video in an equirectangular format, and "View", which calculates the average score after cutting the video into several individual views and evaluating each.

Our IaaW-IM outperforms the baseline FETA across most metrics, demonstrating superior spatial-temporal quality. Temporal metrics averaged across views are higher than overall due to motion discontinuities introduced by splits in the equirectangular format. Aesthetic quality is lower when averaged per view, as certain angles (e.g., top, bottom) naturally lack visual appeal (e.g., sky, floor). VTC-View scores are lower than VTC-All because some view-specific videos inadequately capture the full prompt, reducing alignment.

In continuation model comparisons, our IaaW-IM outperforms the baseline models 360I2V and 4K4DGen [22] across most metrics, indicating stronger spatial and temporal modeling. 4K4DGen is an image animation baseline method capable of processing high-resolution images up to 4K. As this method does not involve text, the metric for view-text consistency is omitted here. Our IaaW method surpasses the 4K4DGen baseline by offering view change, world continuation, and language control, in addition to producing superior video generation effects. This comprehensive set of features highlights the advanced capabilities of our IaaW framework. Temporal and spatial metrics trends mirror those in initialization models, but SC-View is lower than SC-All due to reduced uncertainty when the full panoramic image is available. Temporal metrics surpass those of initialization models as full panoramic input offers richer context than single-view inputs. Slightly lower spatial scores stem from decreased diversity and aesthetic richness when multi-view information is provided.

For whole-process comparison, our IaaW-IM+CM surpasses the baseline FETA+360I2V across all metrics, demonstrating enhanced temporal consistency and spatial quality. Specifically, the overall results are relatively lower than those of the continuation stage. This difference arises because the combination of the initialization and continuation stages makes achieving temporal smoothness more challenging. Since each stage has its own specifications, the overall result evaluates the concatenated videos to achieve a balanced performance metric. By effectively integrating initialization and continuation models, our pipeline generates visually consistent results, whereas the baseline exhibits fragmentation between two stages, leading to inferior performance.

| Model | SC-View | SC-All | MS-View | MS-All | AQ-View | AQ-All | VTC-View | VTC-All |
|---|---|---|---|---|---|---|---|---|
| FETA | 89.4 | 86.8 | 98.1 | 98.3 | 49.9 | 56.7 | 3.19 | 3.93 |
| IaaW-IM | 91.8 | 88.2 | 99.0 | 98.9 | 55.9 | 59.8 | 3.72 | 4.00 |
| 360I2V | 92.5 | 94.8 | 98.9 | 98.7 | 49.5 | 55.0 | 3.25 | 3.89 |
| 4K4DGen | 94.1 | 95.1 | 99.2 | 98.8 | 46.0 | 53.4 | - | - |
| IaaW-CM | 95.8 | 97.2 | 99.3 | 99.2 | 49.7 | 55.7 | 3.26 | 3.90 |
| FETA+360I2V | 81.0 | 88.7 | 98.8 | 98.7 | 50.1 | 55.9 | 3.39 | 3.93 |
| IaaW-IM+CM | 91.0 | 90.3 | 99.1 | 99.1 | 50.5 | 57.5 | 3.50 | 3.94 |

Table 1: Analysis of video generation results of our method and several baselines. SC, MS, AQ and VTC represent subject consistency, motion smoothness, aesthetic quality, and video-text consistency respectively, and for all of these metrics, higher scores are better. Postfix "View" means the numbers are calculated across different views and "All" means the numbers are calculated as a whole. 4K4DGen can generate 4096×2048 resolution video, while for comparison, we include the results using the $2048 \times 1024$ video resolution here.

| Model | SC-View | SC-All | MS-View | MS-All | AQ-View | AQ-All | VTC-View | VTC-All |
|---|---|---|---|---|---|---|---|---|
| IaaW-IM | 91.8 | 88.2 | 99.0 | 98.9 | 55.9 | 59.8 | 3.72 | 4.00 |
| IaaW-IM *w.o.* 3D SphereRoPE | 86.3 | 83.7 | 98.1 | 97.9 | 48.8 | 56.8 | 3.17 | 3.97 |
| IaaW-IM *w.o.* MultiViewComp | 91.2 | 86.6 | 99.0 | 98.9 | 49.5 | 59.9 | 3.24 | 3.95 |

Table 2: Ablation study on our world initialization model IaaW-IM.

## 4.4 Ablation Study

We conduct an ablation study on world initialization components in Table 2. Removing the 3D Spherical RoPE consistently degrades performance both in spatial and temporal metrics, especially for view-based metrics. This degradation is primarily due to spatial distortions, resulting in unsmooth motion and scene deformation. Excluding the Multi-View Composition module reduces VTC-View and AQ-View, as it limits the model's ability to capture view-specific textual cues and leads to a loss of visual quality in separate views. Temporal metrics remain relatively stable, since this module mainly enhances diversity rather than motion smoothness. The slight drop in VTC-All suggests the model still generates prompt-aligned content overall, as neither component directly influences overall textual understanding in video generation.

## 5 Limitations and Social Impact

IaaW excels at generating panoramic world from a single image but struggles with maintaining temporal consistency over very long durations. Specifically, IaaW-CM conditions on the most recent video chunk rather than the full video history, which can lead to a loss of coherence during super long-term video continuations. This long-term consistency presents a key challenge not only for IaaW but also for the broader field of video generation, which we leave as future work.

From a societal perspective, IaaW empowers content creation across VR/AR and gaming, potentially opening new avenues for immersive interactive experiences. However, it also introduces risks related to misinformation and visual deception, which may undermine trust in visual media. Implementing robust safeguards is essential to mitigate potential misuse and ensure responsible use.

## 6 Conclusion

We introduce Image as a World (IaaW), a novel framework for generating expandable, user-controllable panoramic world from a single image, which comprises three critical components: world initialization, world exploration, and world continuation. We design visual world models equipped with 3D spherical RoPE and multi-view composition, and two variants of which, IaaW-IM and IaaW-CM, tackle world initialization and continuation, respectively. Extensive experiments validate the effectiveness of our approach, demonstrating high fidelity, controllability, and scalability across diverse scenarios. Our work opens new potential for one-shot visual world generation in applications such as gaming and virtual reality, setting the stage for future research in generating interactive visual worlds.

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

## A    Theory Analysis

Rotation equivariance in this paper means that rotating the spherical coordinates by an angle results in an identical rotation within the embedding space of the positional encoding—so the model perceives a consistently "shifted" feature without altering the underlying structure. Therefore, to establish rotation-equivariance, it suffices to show that any additive shift in longitude $\phi$, latitude $\theta$, or time $t$ (as defined in E Eqs. (1) and (2)) induces an equivalent orthogonal rotation within the corresponding subspace of the encoding. We first recall the definitions from Eqs. (1) and (2):

$$\theta = \frac{\pi}{2}(\frac{2y}{H} - 1), \phi = \pi(\frac{2x}{W} - 1), \tau = 2\pi t,$$

and for each frequency index $d = 0, 1, ..., N - 1$,

$$\text{RoPE}_{x,y,t,d} = \begin{bmatrix} \cos(2^d\theta) \cdot \cos(2^d\phi) \cdot \cos(2^d\tau) \\ \sin(2^d\theta) \cdot \cos(2^d\phi) \cdot \cos(2^d\tau) \\ \cos(2^d\theta) \cdot \sin(2^d\phi) \cdot \cos(2^d\tau) \\ \sin(2^d\theta) \cdot \sin(2^d\phi) \cdot \cos(2^d\tau) \\ \cos(2^d\theta) \cdot \cos(2^d\phi) \cdot \sin(2^d\tau) \\ \sin(2^d\theta) \cdot \cos(2^d\phi) \cdot \sin(2^d\tau) \\ \cos(2^d\theta) \cdot \sin(2^d\phi) \cdot \sin(2^d\tau) \\ \sin(2^d\theta) \cdot \sin(2^d\phi) \cdot \sin(2^d\tau) \end{bmatrix} \in \mathbb{R}^8$$

The full positional embedding is the concatenation of all frequencies as

$$f(x, y, t) = [\text{RoPE}_{x,y,t,0}||\text{RoPE}_{x,y,t,1}||\text{RoPE}_{x,y,t,2}||...] \in \mathbb{R}^{8N}$$

To prove that $f$ is equivariant under any rotation $R$ (acting on $(x, y)$ via the induced changes in $(\theta, \phi)$) and any time shift $t \to t + \Delta t$, it suffices to show equivariance dimension-wise. For clarity, we use a rotation in the longitude dimension $\phi$ as an example. We begin by expressing each 8-dimensional block $\text{RoPE}_{x,y,t,d}$ in terms of complex expoential $c_\theta = e^{i2^d\theta}$, $c_\phi = e^{i2^d\phi}$, $c_\tau = e^{i2^d\tau}$, so that each entry of $\text{RoPE}_{x,y,t,d}$ is the real and imaginary part of a product $c_\theta^\alpha c_\phi^\beta c_\tau^\gamma$ with $\alpha, \beta, \gamma \in \{0, 1\}$. In the complex domain, an additive shift, i.e. $\phi \to \phi + \Delta\phi$, corresponds to multipication by a phase factor $e^{i2^d\Delta\phi}$. This implies that each sin/cos pair involving $\phi$, i.e., $(\cos(2^d\Delta\phi), \sin(2^d\Delta\phi))$, lies in a 2-dimensional subspace rotated by a 2×2 orthoganal matrix

$$R_d^\phi(2^d\Delta\phi) = \begin{pmatrix} \cos(2^d\Delta\phi) & -\sin(2^d\Delta\phi) \\ \sin(2^d\Delta\phi) & \cos(2^d\Delta\phi) \end{pmatrix}$$

This rotation leaves the $\theta$ and $t$ components unaffected expect for being scaled by fixed multiplicative factors, thus perserving equivariant within the full 8-dimension embedding. There are four such pairs in $\text{RoPE}_{x,y,t,d}$, which results in an 8x8 block-diagonal orthogonal matrix $T_d^\phi$. Stacking the matrices across all the frequencies $d = 0, ..., N - 1$ yields a global orthogonal matrix

$$T^\phi = diag(T_0^\phi, ..., T_{N-1}^\phi) \in \mathbb{R}^{8N \times 8N}$$

which satisfies

$$f(R^\phi \cdot (x, y, t)) = T^\phi \cdot f(x, y, t)$$

Here $R^\phi$ denotes the rotation applied in longitude. The same reasoning applies to $\theta$ and time $\tau$, where additive shifts similarly induce orthogonal transformations within their respective subspaces. Therefore, spatial rotations and temporal shifts are exactly mirrored by orthogonal rotations in the embedding space, confirming that the 3D spherical RoPE is rotation-equivariant.

## B    Efficiency Analysis

According to Table A, our method achieves inference times comparable to both the base foundation model (despite operating at higher resolution) and to prior work such as 4K4DGen (while generating more frames). Currently, each generation step (covering ~3–5 seconds of video) takes approximately 10–20 minutes on a single A100 GPU. This latency stems from the high output resolution (2048×1024) and the large size of the backbone models.

| Model | Resolution | Inference Time | Inference Memory Usage |
|---|---|---|---|
| CogVideoX1.5-5B-I2V(base) | 1360×768×81 | $\sim 16$ min | $\geq 9$ GB |
| 4k4DGen* | 2048×1024×14 | $\sim 16$ min | $\geq 12$ GB |
| IaaW-IM | 2048×1024×49 | $\sim (17 \times n)$ min | $\geq (12 \times n)$ GB |
| IaaW-CM | 2048×1024×49 | $\sim 17$ min | $\geq 12$ GB |

Table A: Analysis of video generation efficiency of our method and several baselines.

As with most generative systems, there exists a trade-off between generation quality (e.g., resolution) and latency. In this work, we prioritize generation quality, though we also discuss various optimization techniques that could accelerate inference as follow.

- Multi-GPU Deployment: Utilizing FSDP or DeepSpeed Ulysses to parallelize inference across GPUs.
- Model Compression and Acceleration: Techniques such as increasing the VAE encoding granularity—e.g., encoding larger spatial chunks as in LTX-Video [12]—can significantly reduce computational cost.
- Efficient Attention Mechanisms: Incorporating architectural improvements such as Pyramid Attention Broadcast (PAB) [46] can help accelerate DiT-based video generation.
- Autoregressive Frame Scheduling: Reducing the number of frames generated at each step and progressively extending sequences in an autoregressive fashion (as explored in recent work like AAPT [24]) may enable near-real-time inference with minimal quality compromise.

## C   Failure Case Analysis

In terms of performance on highly complex scenes, such as urban street views, we find that our model is less reliable compared to natural or less cluttered environments. For instance, in one case involving an aerial view of a busy urban street with numerous cars, the generated video exhibited unnatural behavior: some cars remained static while others moved in inconsistent or physically implausible directions. There are two main contributing factors:

1. Model Capacity: The base models we built on (including CogVideoX and other comparable open-source video diffusion models) struggle to robustly handle scenarios with multiple independently moving objects, which exceeds the temporal modeling capacity of existing models.
2. Training Data Bias: To ensure visual stability, we filtered out videos with significant camera shake—many of which were hand-held recordings containing dense motion and multiple objects. As a result, the training set is less representative of such complex dynamic environments, which impacts generalization.

We have also conducted a preliminary failure case analysis and identified several common failure modes

1. Complex Motion or Scene Crowding: Scenes with a high density of independently moving objects (e.g., vehicles, pedestrians) often lead to degraded performance. The base model's capacity to process multiple interacting objects attention is limited, resulting in static or erratically moving elements.
2. Human Actions: When humans are present in the scene, motion may be unrealistic or static. This is partly due to the difficulty of modeling articulated human motion in video diffusion models, and further exacerbated in 360-degree video due to varying perspective motion distortions across the sphere.
3. Unsuited Input Scenarios: Close-up views of objects, animals, or plants often result in implausible generations—such as oversized elements or distorted layouts—due to the egocentric nature of 360-degree video. When the initial view covers a very narrow or zoomed-in area, the extrapolated scene tends to resemble a "Lilliput effect", where everything appears disproportionately large. We find that the IaaW works best for wide-field scenes like indoor rooms, landscapes, or aerial views, where surrounding context is available.

| Parameters | Value |
| --- | --- |
| Video Height | 1024 |
| Video Width | 2048 |
| Video Frames | 49 |
| Batch Size | 4 |
| Mixed Precision | bf16 |
| Optimizer | AdamW |
| Optimizer Betas | (0.9, 0.95) |
| Optimizer Weight Decay | 1e-4 |
| Learning Rate | 2e-5 |
| Warmup Steps | 100 |
| Inference Steps | 25 |
| Frame Per Second | 16 |
| Masks Number | 6 |

Table B: Hyperparameters setting of our experiments.

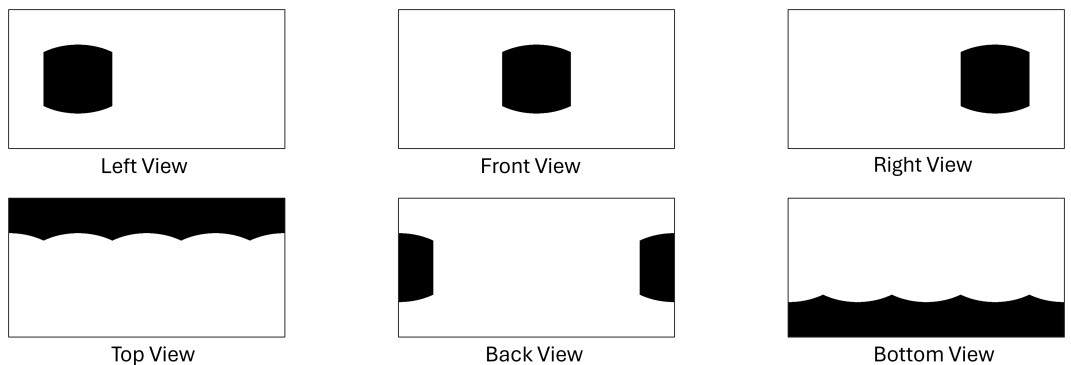

Figure A: Different view masks setup in our method.

## D  Experiments Setup

We present the hyperparameters used in our experiment in Table B. The same hyperparameters are applied to both IaaW-IM and IaaW-CM, and our pipeline is built upon the CogVideoX codebase.

We also present our masks setup in Fig. A, which contains six perspectives of one panoramic image/video and these can together seamlessly reconstruct the full panoramic scene.

## E  More Visualization Results

We present a qualitative visualization comparison between WonderWorld [44] and our IaaW in Fig. B. The results find that our method generate a panoramic dynamic world instead of a single static 3D world scene compared with WonderWorld.

We present additional visualization results on complex scenes and indoor scenes in Fig. C, which demonstrate that our method exhibits significant diversity across various scenarios.

We also present additional visualization results of our world initialization in Fig. D.

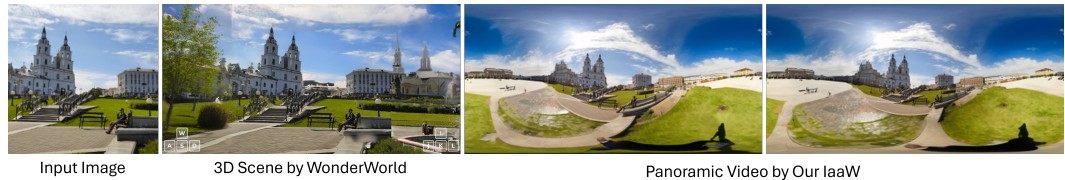

Input Image          3D Scene by WonderWorld          Panoramic Video by Our IaaW

Figure B: Qualitative visualization comparison between WonderWorld and our IaaW.

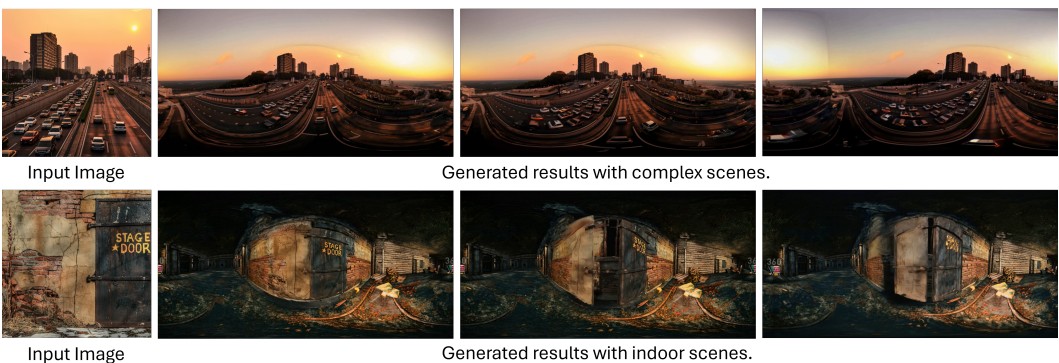

Input Image          Generated results with complex scenes.

Input Image          Generated results with indoor scenes.

Figure C: Qualitative visualization on complex scenes and indoor scenes.

Input Image          World Initialization Results

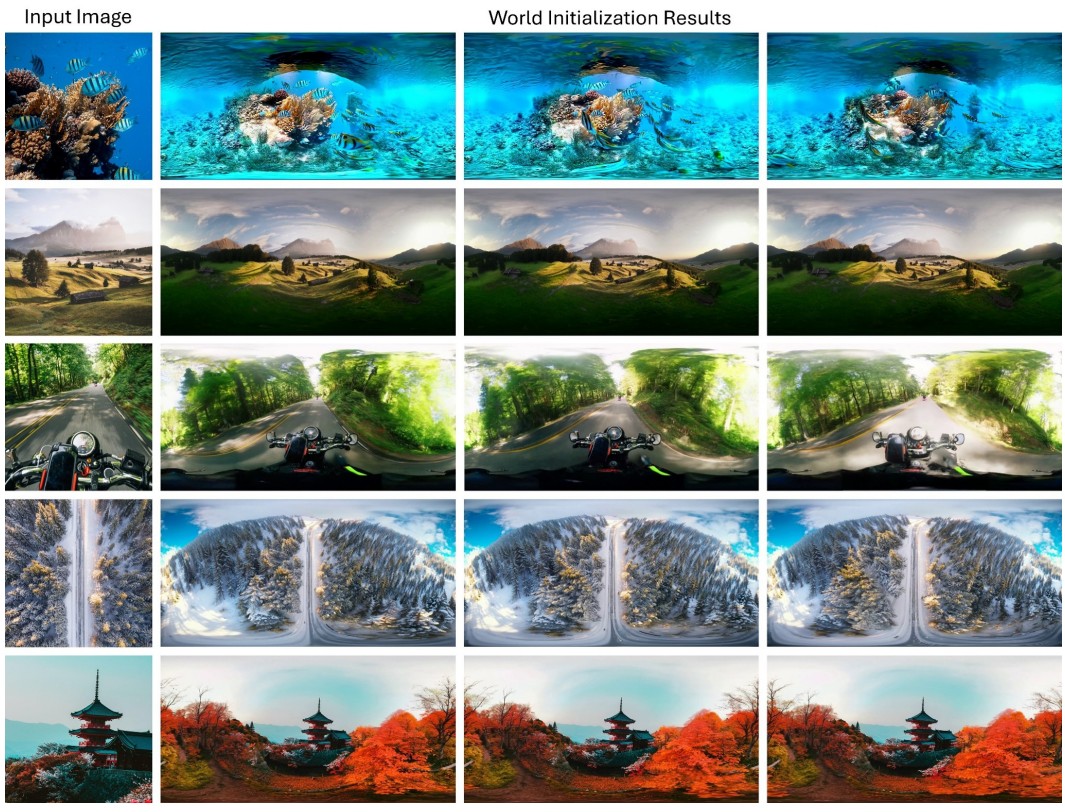

Figure D: More visualization results of world initialization.

