# OpenReview forum: "Image as a World: Generating Interactive World from Single Image via Panoramic Video Generation"
_NeurIPS.cc/2025/Conference — NeurIPS 2025 poster_

### Official Review · Reviewer_4SKk · 2025-06-16

**Clarity:** 2
**Significance:** 2
**Originality:** 2
**Rating:** 2
**Confidence:** 4

**Summary:**

This paper introduces a novel framework called Image as a World (IaaW) , which generates interactive, panoramic 360° video worlds from a single input image. The key innovation lies in its ability to not only generate high-quality panoramic videos but also allow user-controlled viewpoint navigation and temporal continuation , enabling the synthesized world to evolve indefinitely while maintaining spatial and temporal coherence. It makes a contribution to the field of generative modeling by framing panoramic video synthesis as a visual world modeling task , bridging the gap between static image generation and dynamic, interactive environment creation.

**Questions:**

1) How to theoretically demonstrate the effectiveness of the method proposed in the paper?

2) Can you provide inference time statistics (per frame or per segment) and discuss how the three‑stage pipeline might be optimized for real‑time or near‑real‑time applications?

3) How well does IaaW handle highly complex or cluttered scenes (e.g., urban street views) compared to the mostly outdoor/natural examples shown? Could you include a failure case analysis?

**Ethical Concerns:**

["NO or VERY MINOR ethics concerns only"]

**Final Justification:**

I have carefully read the authors' responses to my concerns and comments, as well as the interactions between the authors and all reviewers. I think some of reviewer concerns have not been satisfactorily resolved yet. I think this paper still needs further improvement by resolving these unresolved concerns. Therefore, in my opinion, it does not yet meet the standards required by the conference.

**Limitations:**

The authors acknowledge the key limitation that the continuation model conditions only on the most recent video chunk, causing coherence drift in very long sequences. Additionally, the requirement for multi‑GPU, multi‑week finetuning and high‑resolution datasets limits the method’s practicality. A more thorough discussion of runtime, memory footprint, and potential mitigations (e.g., chunked encoding, model distillation) would strengthen the work.

**Paper Formatting Concerns:**

No major formatting issues noted.

**Quality:**

2

**Strengths And Weaknesses:**

Strengths:
1) Tackles the challenging task of producing a fully interactive, temporally extensible panoramic world from just a single view.
2) The 3D spherical RoPE is a neat extension of rotary embeddings to equirectangular video, and the multi‑view composition module offers fine‑grained control over scene synthesis.
3) Uses diverse quality, temporal, and semantic metrics, with clear improvements over several strong baselines. Detailed ablation studies demonstrate the impact of each proposed component.

Weaknesses
1) The paper contributes primarily through engineering and system design rather than theoretical innovation. While practical frameworks are valuable, this limits its broader scientific impact.
2) Generation of 2048×1024 videos in three stages on multiple A100 GPUs over several weeks limits practical applicability; no runtime or resource‑efficiency analysis is provided.
3) The continuation stage conditions only on the most recent chunk, leading to drift over very long sequences, as admitted in the Limitations section.

---

> ### Author Rebuttal · Authors · 2025-07-31
>
> We greatly appreciate the reviewer’s recognition of our contributions. Below, we provide detailed responses to the reviewer’s concerns and questions.
>
> > The paper contributes primarily through engineering and system design rather than theoretical innovation. While practical frameworks are valuable, this limits its broader scientific impact. How to theoretically demonstrate the effectiveness of the method proposed in the paper?
>
> While our panoramic video generation pipeline may appear engineering-oriented at first glance, each stage is carefully designed based on theoretically analysis and reflects an optimal solution to its specific subproblem. Although our work does not claim purely theoretical innovation, the overall system is built upon principled reasoning rather than engineering convenience or performance tuning, addressing the fundamental ambiguities of visual world generation.
>
> 1. Stage 1: Single-view to spherical dynamics
>
>     This stage tackles the underdetermined task of reconstructing 360° motion from a single static image. We formulate it as an ill-posed inverse problem and reduce ambiguity through strong priors: spherical RoPE for rotational symmetry and multi-view composition for consistency and diversity. These components constrain the solution space and enable plausible world reconstruction from sparse input.
>
> 2. Stage 2: View-consistent rotation
>
>     This stage addresses the challenge of generating view-consistent panoramic video under camera motion. We design a group-equivariant rotation module that ensures latent representations transform coherently with spherical rotations. By leveraging equivariance theory, we guarantee that changes in viewpoint result in structured transformations in the embedding space. This property is validated both theoretically and empirically.
>
> 3. Stage 3: Time-consistency generation
>
>     This stage ensures long-range temporal consistency in panoramic video generation. By conditioning on embeddings from previous chunks, we reduce visual drift and maintain coherent scene dynamics over time. Our design leverages diffusion convergence insights to stabilize generation across successive segments.
>
> These stages form a principled and interpretable solution to the problem of extending a static view into an immersive, controllable, and coherent video world—bridging perception, geometry, and generation under a unified theoretical framework. We really appreciate this comment and will include corresponding discussion to provide additional insights in the revised version.
>
> > Generation of 2048×1024 videos in three stages on multiple A100 GPUs over several weeks limits practical applicability;
>
> Thank you for raising this important concern. While training was indeed resource-intensive, we believe the practical applicability remains reasonable for two key reasons:
> 1. The multi-week duration refers only to model training. Once trained, only the inference stage is required for user applications, which is significantly faster and more scalable.
> 2. The long training time was primarily due to limited hardware availability. With access to larger compute clusters, the process can be accelerated to days or even hours. Moreover, several inference-time optimizations can further improve runtime and deployment efficiency, which will be discussed in the next part.
>
> > No runtime or resource‑efficiency analysis is provided. Can you provide inference time statistics (per frame or per segment) and discuss how the three‑stage pipeline might be optimized for real‑time or near‑real‑time applications?
>
> Thank you for raising this important point. We agree that runtime and efficiency are critical for assessing practical applicability, and we will include detailed analysis in the revised version. We present our inference time statistics as below.
>
> | Model                     | Resolution   | Inference Time       | Inference Memory Usage    |
> | ------------------------- | ------------ | -------------------- | ------------------------- |
> | CogVideoX1.5-5B-I2V(base) | 1360\*768\*81  | ~16 min              | $\ge $9GB               |
> | 4k4DGen\*                  | 2048\*1024\*14 | ~16 min              | $\ge $12GB              |
> | IaaW-IM                   | 2048\*1024\*49 | ~(17*n) min n=\#views | $\ge $(12*n)GB n=\#views |
> | IaaW-CM                   | 2048\*1024\*49 | ~17 min              | $\ge $12GB              |
>
> $\small{\text{4k4DGen with \* here represents it can generate 4096*2048 resolution video, while for comparison, we include the 2048\*1024 results here.}}$
>
> It can be observed that our method achieves inference times comparable to both the base foundation model (despite operating at higher resolution) and to prior work such as 4K4DGen (while generating more frames). Currently, each generation step (covering ~3–5 seconds of video) takes approximately 10–20 minutes on a single A100 GPU. This latency stems from the high output resolution (2048×1024) and the large size of the backbone models. Current real-time methods often use lower resolution and smaller models, which trade off generation quality compared to our approach.
>
> We acknowledge that this inference time limits real-time deployment and we are actively exploring several optimization directions:
> 1. Multi-GPU Deployment: Utilizing FSDP or DeepSpeed Ulysses to parallelize inference across GPUs.
> 2. Model Compression and Acceleration: Techniques such as increasing the VAE encoding granularity—e.g., encoding larger spatial chunks as in LTX-Video[1]—can significantly reduce computational cost.
> 3. Efficient Attention Mechanisms: Incorporating architectural improvements such as Pyramid Attention Broadcast (PAB) [2] can help accelerate DiT-based video generation.
> 4. Autoregressive Frame Scheduling: Reducing the number of frames generated at each step and progressively extending sequences in an autoregressive fashion (as explored in recent work like AAPT[3]) may enable near-real-time inference with minimal quality compromise.
>
> [1]:HaCohen, Yoav, et al. "Ltx-video: Realtime video latent diffusion." arXiv preprint arXiv:2501.00103 (2024).
>
> [2]:Zhao, Xuanlei, et al. "Real-time video generation with pyramid attention broadcast." arXiv preprint arXiv:2408.12588 (2024).
>
> [3]:Lin S, Yang C, He H, et al. Autoregressive Adversarial Post-Training for Real-Time Interactive Video Generation[J]. arXiv preprint arXiv:2506.09350, 2025.
>
> > The continuation stage conditions only on the most recent chunk, leading to drift over very long sequences, as admitted in the Limitations section.
>
> Thank you for your valuable feedback. As acknowledged in the Limitations section, our current continuation stage conditions only on the most recent chunk, which can lead to drift over very long sequences. This design choice is primarily due to memory constraints, which prevent us from incorporating a longer history during generation. However, we are actively exploring strategies to better utilize historical frames—for example, by compressing them or designing mechanisms that allow us to integrate as much historical context as possible within available memory. We believe such improvements can significantly enhance long-term consistency in future iterations.
>
> > How well does IaaW handle highly complex or cluttered scenes (e.g., urban street views) compared to the mostly outdoor/natural examples shown? Could you include a failure case analysis?
>
> Thank you for this important question. We are unable to provide additional visualizations or failure case examples during the rebuttal phase, but we will include a detailed section with such cases in the revised version of the paper and supplementary materials.
>
> In terms of performance on highly complex scenes, such as urban street views, we find that our model is less reliable compared to natural or less cluttered environments. For instance, in one case involving an aerial view of a busy urban street with numerous cars, the generated video exhibited unnatural behavior: some cars remained static while others moved in inconsistent or physically implausible directions. These limitations are primarily due to the challenges of modeling complex, multi-object motion—an area where current video diffusion backbones (including CogVideoX and comparable open-source models) still struggle, due to limited temporal modeling capacity for dense and heterogeneous motion.
>
> We have conducted a preliminary failure case analysis and identified several common failure modes:
> 1. Complex Motion or Scene Crowding: Scenes with a high density of independently moving objects (e.g., vehicles, pedestrians) often lead to degraded performance. The base model's capacity to process multiple interacting objects attention is limited, resulting in static or erratically moving elements.
> 2. Human Actions: When humans are present in the scene, motion may be unrealistic or static. This is partly due to the difficulty of modeling articulated human motion in video diffusion models, and further exacerbated in 360-degree video due to varying perspective motion distortions across the sphere.
> 3. Unsuited Input Scenarios: Close-up views of objects, animals, or plants often result in implausible generations—such as oversized elements or distorted layouts—due to the egocentric nature of 360° video. When the initial view covers a very narrow or zoomed-in area, the extrapolated scene tends to resemble a “Lilliput effect,” where everything appears disproportionately large. We find that the IaaW works best for wide-field scenes like indoor rooms, landscapes, or aerial views, where surrounding context is available.

---

> > ### Comment · Reviewer_4SKk · 2025-08-07
> >
> > Thanks for the authors' response, which addresses part of my concerns. Based on the authors' feedback and other reviewers' comments, I will consider adjusting my rating. Thank you.

---

> > > ### Author Response · Authors · 2025-08-07
> > >
> > > We’re glad our rebuttal helped clarify some of your concerns, and we appreciate your consideration in adjusting your rating. Thank you again for your thoughtful review.

---

### Official Review · Reviewer_8ork · 2025-07-02

**Clarity:** 1
**Significance:** 3
**Originality:** 3
**Rating:** 4
**Confidence:** 4

**Summary:**

This paper proposes a framework (IaaW) for a panaromic video generation and continuation from a single image. An image is taken as input, and a panaromic video is made from it (IaaW-IM). Then, this panaromic scene can be explored by turning around etc., which is realized using a unified spherical RoPE. Then, the panaromic scene is extended as a video from the exploration point (IaaW-CM). Experiments are conducted on a CogVideoX model finetuned on the relevant data. To help with the generation, a VLM is also trained to generate the right prompts relevant to the task.

**Questions:**

"Unlike traditional world models that focus on action-conditioned prediction, our approach synthesizes immersive scenes that respond to user-specified actions, enabling both free-form viewpoint control and continuous scene expansion."
Not sure what the difference is between "action-conditioned prediction" and "synthesis of immersive scenes that respond to user-specified actions", both seem to be the same thing.

Fig.3 and 3.1.1. text mismatch needs to be explained.

It is quite unclear what the masks or the view prompts are.

It is unclear why the VLM itself is so important for this task.

In the supplementary material, sample-*-2WorldExploration.mp4 and sample-*.mp4 seem to be empty flies, they just contain a blank green video.

**Ethical Concerns:**

["NO or VERY MINOR ethics concerns only"]

**Final Justification:**

The authors have provided a good point-by-point rebuttal to the review, hence I decided to increase my rating of the submission.

**Limitations:**

yes

**Paper Formatting Concerns:**

Line 29: explodable --> explorable

**Quality:**

2

**Strengths And Weaknesses:**

The paper takes on a challenging task and tries to figure out a useful framework to solve it.

It chooses to finetune a video-based diffusion model to first generate a panaromic video, thereby solving the 4D world generation task. A panaromic view can be considered 3D, and making it a video effectively renders it 4D, hence it is a world. This is an innovative approach, picking up popularity in recent times. However, since it uses a diffusion model in the backend, it may not be close to real time yet, at least not for initialization.

Fig.3 is quite unaligned with the corresponding text in 3.1.1. The text talks about "parallel paths" and "localized 3D full attention" suggesting that the original MMDiT is unchanged, and a gated MMDiT with the vision component being concatenated with view features. But Fig.3 indicates attention on all fields : text, vision, and vision concatenated with few features. This needs to be clarified.

Also, it is quite unclear what the masks or the view prompts are. Relevant mention is in sections 3.1.1 and 3.2.1, but they haven't been explained yet.

Then, to explore the world, the paper proposes a unified spherical position embedding. This is very innovative and relevant for this task. It is to be noted that this exploration is in a static space, not dynamic : a static world is explored, then continued as a video. Nevertheless, the use of spherical embedding for panorama and extension to time for video is a nice adaptation of both techniques for this task.

Section 3.2.1 and Fig. 4 show IaaW-VLM takes the world video as input, but the original task and Fig. 2 start from only a single image. It is unclear how the first video itself is given as input to IaaW-VLM before it is generated by IaaW-IM.

It is unclear why the VLM itself is so important for this task. Maybe more clarity can be achieved by knowing the use of view prompts and masks.

Overall, the paper presents the problem and a simplified summary of the method very well, but the figures and methods section need improvement for clarity.

The supplementary material has very useful video examples of each stage of this task. However, the exploration part is a blank green video in each example.

---

> ### Author Rebuttal · Authors · 2025-07-31
>
> We greatly appreciate the reviewer's recognition of the challenging nature of 4D world generation and the innovative value of our approach. We appreciate your recognition of our use of panoramic video as a 4D representation, the novelty of our unified spherical position embedding, and the relevance of adapting panoramic spatial modeling to temporal progression. Below, we provide detailed responses to the reviewer’s concerns and questions.
>
> > Fig.3 is quite unaligned with the corresponding text in 3.1.1. The text talks about "parallel paths" and "localized 3D full attention" suggesting that the original MMDiT is unchanged, and a gated MMDiT with the vision component being concatenated with view features. But Fig.3 indicates attention on all fields : text, vision, and vision concatenated with few features. This needs to be clarified.
>
> In fact, the MMDiT block is indeed modified in our approach. As shown in Fig. 3, we introduce an additional view-aware branch, highlighted by the grey background. This newly added branch operates in parallel with the original text + vision stream. Both the original and the new branches apply 3D full attention independently. The original branch processes the standard text + vision input, while the added branch incorporates view features by taking auxiliary view prompts, applying corresponding spatial masks, and performing 3D full attention over the masked vision regions. The outputs from both branches are then fused through a gated mechanism to produce the final output. We will revise Section 3.1.1 to explicitly state that the MMDiT architecture is modified with an additional view-aware path and clarify the roles of each attention path to better align with Fig.3.
>
> > Also, it is quite unclear what the masks or the view prompts are. Relevant mention is in sections 3.1.1 and 3.2.1, but they haven't been explained yet.
>
> Thank you for pointing this out. We agree that our current version lacks an explicit and clear explanation of the view masks and view prompts, and we will address this in the revised manuscript. In the World Initialization stage, we introduce a multi-view composition mechanism to enhance spatial diversity and controllability. This mechanism takes two inputs:
> - View masks: binary masks that correspond to user-specified predefined views (e.g., front, left, top). The types of masks are illustrated in supplementary material.
> - View prompts: natural language prompts aligned with each masked region generated by our IaaW-VLM, providing textual guidance for content generation from that specific viewpoint.
>
> This setup allows the system to generate a more diverse and spatially view-aware initial world. We will revise Sections 3.1.1 and 3.2.1 to explain this clearer.
>
> > Section 3.2.1 and Fig. 4 show IaaW-VLM takes the world video as input, but the original task and Fig. 2 start from only a single image. It is unclear how the first video itself is given as input to IaaW-VLM before it is generated by IaaW-IM.
>
> Thank you for pointing this out and we appreciate the opportunity to clarify. The IaaW-VLM module is designed to handle two distinct scenarios, as illustrated by the two input-output pathways in Fig. 4. It is employed in both the World Initialization and World Continuation stages, taking an image and a video as input, respectively. Specifically, IaaW-VLM is fine-tuned for two types of inputs: (1) an image input used during the World Initialization stage, and (2) a video input used during the World Continuation stage. In the initialization stage, the original input image is the starting point. IaaW-IM first processes this image to produce an initial equirectangular video prompt $P\_1$, as well as a set of optional view prompts $P\_v$ to support multi-view composition. In the continuation stage, IaaW-VLM then takes the previously generated video as input to produce the next video prompt for further progression. We will clarify this distinction more explicitly and split the figure for the two stages in the revised version.
>
> > It is unclear why the VLM itself is so important for this task.
>
> Thank you for pointing this out and we appreciate the opportunity to clarify. VLM used in our method plays a distinct but crucial role in providing textual information across world initialization stage and world continuation stage.
> In the World Initialization stage, the VLM takes a single-view input image and generates:
> - A holistic video prompt capturing the overall scene evolution.
> - Optional view prompts, each aligned with a predefined view mask (e.g., front, left), enabling multi-view composition with spatial grounding.
>
> In the World Continuation stage, the VLM conditions on the previously generated (and rotated) video and produces the next-step prompt, guiding coherent temporal progression.
>
> By serving as the bridge between visual context and prompt-based video generation, the VLM allows our system to automatically create an extensible and interactive 360-degree world from just a single image. We will revise the manuscript to clarify this mechanism and its importance.
>
> > The supplementary material has very useful video examples of each stage of this task. However, the exploration part is a blank green video in each example.
>
> Thank you for your feedback. We have double-checked the supplementary material and verified that all videos, including the exploration stage, play correctly on our end, includes Photos, Media Player, and PotPlayer in Windows 11.
>
> > "Unlike traditional world models that focus on action-conditioned prediction, our approach synthesizes immersive scenes that respond to user-specified actions, enabling both free-form viewpoint control and continuous scene expansion." Not sure what the difference is between "action-conditioned prediction" and "synthesis of immersive scenes that respond to user-specified actions", both seem to be the same thing.
>
> Thank you for the feedback. Our intention here is to highlight a key distinction: traditional world models typically predict the next state conditioned on actions—i.e., they model action-to-state transitions step by step. In contrast, once our immersive world is synthesized, users can explore it freely via arbitrary viewpoint changes—without requiring step-by-step action conditioning. This is enabled by the inherently 360-degree, rotatable nature of the generated video, and shown in demo video of supplemental materials. We will revise the sentence to better clarify this distinction.

---

> > ### Comment · Reviewer_8ork · 2025-08-05
> > **Thank you**
> >
> > Thank you for the point-by-point response to the review. I will take this into consideration and update my score accordingly.

---

> > > ### Author Response · Authors · 2025-08-06
> > >
> > > We sincerely thank you again for your thoughtful review and for acknowledging the novelty and contributions of our work—particularly in addressing the challenging problem of 4D world generation and introducing the unified spherical position embedding for panoramic scene modeling and temporal extension.
> > >
> > > We hope that our detailed rebuttal has helped clarify the technical aspects you raised. Do you have any further questions or concerns? We would be more than happy to elaborate. We truly value this discussion period and greatly appreciate your time and thoughtful feedback throughout the review and discussion process.

---

### Official Review · Reviewer_fetS · 2025-07-02

**Clarity:** 3
**Significance:** 3
**Originality:** 3
**Rating:** 4
**Confidence:** 4

**Summary:**

IaaW presents a unified framework that generates controllable and temporally consistent 360° videos from a single image. It integrates world initialization, viewpoint exploration, and temporal continuation, enhanced by diffusion models, spherical 3D encoding, and a fine-tuned vision-language model for improved semantic control.

**Questions:**

Kindly refer to the [Weaknesses].

**Ethical Concerns:**

["NO or VERY MINOR ethics concerns only"]

**Final Justification:**

The authors' response and additional experiments have addressed my concerns. Considering the perspectives of the other reviewers, I will maintain my current score.

**Limitations:**

Yes

**Quality:**

3

**Strengths And Weaknesses:**

### Strengths
* This work presents a groundbreaking world generation model capable of producing high-quality dynamic panoramas and supporting interactive experiences from a single image.
* The motivation behind the design of IaaW is well-founded and compelling, with its contributions validated through comparisons against FETA and the 360I2V baseline.
* The paper is well-written and easy to follow.
### Weaknesses
* The section on 3D Spherical RoPE is overly brief. While its effectiveness is demonstrated experimentally, it is recommended to include additional comparisons with other 3D positional encoding methods. Furthermore, a theoretical derivation or proof of its rotation-equivariant properties, as suggested by Equation 2, would be a valuable addition to the appendix.
* More comparative methods should be included. Although there may be no direct baselines for the entire task, fair comparisons can still be made at the "world initialization" and "world continuation" stages with recent panoramic video generation and interactive world synthesis methods (e.g., 4K4DGen, OmniDrag, WonderWorld).
* Efficiency analysis. As an interactive world generation model, the reliance on VLMs and video diffusion raises concerns about inference and interaction latency. It is recommended to include experimental results assessing the model's efficiency.

---

> ### Author Rebuttal · Authors · 2025-07-31
>
> We greatly appreciate the reviewer’s recognition of our contributions. Below, we provide detailed responses to the reviewer’s concerns and questions.
>
> > The section on 3D Spherical RoPE is overly brief. While its effectiveness is demonstrated experimentally, it is recommended to include additional comparisons with other 3D positional encoding methods. Furthermore, a theoretical derivation or proof of its rotation-equivariant properties, as suggested by Equation 2, would be a valuable addition to the appendix.
>
> Thank you for the insightful suggestion. We will include additional 3D positional encoding methods for comparison in the revised manuscript. Our proposed 3D spherical RoPE extends traditional 3D rotary embeddings to 4D by modulating angular and temporal components, which significantly enhances the structure generation and temporal consistency of panoramic video.
>
> We also appreciate the suggestion to provide a theoretical proof of the rotation equivariance of our method. We will include the following proof in the appendix of the revised manuscript.
>
> ### Proof of Rotation Equivariance
>
> Rotation equivariance in this paper means that rotating the spherical coordinates by an angle results in an identical rotation within the embedding space of the positional encoding—so the model perceives a consistently "shifted" feature without altering the underlying structure. Therefore, to establish rotation-equivariance, it suffices to show that any additive shift in longitude $\phi$, latitude $\theta$, or time $t$ (as defined in Equations 1 & 2) induces an equivalent orthogonal rotation within the corresponding subspace of the encoding.
>   We first recall the definitions from Equations 1 & 2:
>   $$\theta = \frac{\pi}{2}(\frac{2y}{H}-1), \phi = \pi(\frac{2x}{W}-1), \tau = 2 \pi t,$$
>   and for each frequency index $d=0,1,...,N-1,$
> $$\text{RoPE}\_{x,y,t,d}=\left[ \begin{matrix}
>     \cos(2^d \theta) \cdot \cos(2^d \phi) \cdot \cos(2^d \tau) \\\\
>     \sin(2^d \theta) \cdot \cos(2^d \phi) \cdot \cos(2^d \tau) \\\\
>     \cos(2^d \theta) \cdot \sin(2^d \phi) \cdot \cos(2^d \tau) \\\\
>     \sin(2^d \theta) \cdot \sin(2^d \phi) \cdot \cos(2^d \tau) \\\\
>     \cos(2^d \theta) \cdot \cos(2^d \phi) \cdot \sin(2^d \tau) \\\\
>     \sin(2^d \theta) \cdot \cos(2^d \phi) \cdot \sin(2^d \tau) \\\\
>     \cos(2^d \theta) \cdot \sin(2^d \phi) \cdot \sin(2^d \tau) \\\\
>     \sin(2^d \theta) \cdot \sin(2^d \phi) \cdot \sin(2^d \tau) \\\\
> \end{matrix}
>   \right] \in \mathbb{R}^8$$
> The full positional embedding is the concatenation of all frequencies as
> $$f(x,y,t)=\left[ \text{RoPE}\_{x,y,t,0} || \text{RoPE}\_{x,y,t,1} || \text{RoPE}\_{x,y,t,2} || ...\right] \in \mathbb{R}^{8N}$$
> To prove that $f$ is equivariant under any rotation $R$ (acting on $(x,y)$ via the induced changes in $(\theta,\phi)$) and any time shift $t\rightarrow t+\Delta t$, it suffices to show equivariance dimension-wise. For clarity, we use a rotation in the longitude dimension $\phi$ as an example.
>
> We begin by expressing each 8-dimensional block $\text{RoPE}\_{x,y,t,d}$ in terms of complex expoential
>  $c_\theta=e^{i2^d\theta}$, $c_\phi=e^{i2^d\phi}$, $c_\tau=e^{i2^d\tau}$,
> so that each entry of $\text{RoPE}\_{x,y,t,d}$ is the real and imaginary part of a product $c^\alpha_\theta c^\beta_\phi c^\gamma_\tau$ with $\alpha,\beta,\gamma \in \\{0,1\\}$.
> In the complex domain, an additive shift, i.e.$\phi \rightarrow \phi+\Delta\phi$, corresponds to multipication by a phase factor $e^{i2^d\Delta\phi}$. This implies that each sin/cos pair involving $\phi$, i.e., $(\cos(2^d \Delta \phi),\sin(2^d \Delta \phi))$, lies in a 2-dimensional subspace rotated by a 2x2 orthoganal matrix $R^\phi\_d(2^d\Delta\phi)=( \begin{matrix}
>     \cos(2^d \Delta \phi) & -\sin(2^d \Delta \phi) \\\\
>     \sin(2^d \Delta \phi) & \cos(2^d \Delta \phi)
> \end{matrix})$.
> This rotation leaves the $\theta$ and $t$ components unaffected expect for being scaled by fixed multiplicative factors, thus perserving equivariant within the full 8-dimension embedding.
>
> There are four such pairs in $\text{RoPE}\_{x,y,t,d}$, which results in an 8x8 block-diagonal orthogonal matrix $T^\phi_d$. Stacking the matrices across all the frequencies $d=0,...,N-1$ yields a global orthogonal matrix
> $T^{\phi}=diag(T^\phi_0,...,T^\phi_{N-1})\in \mathbb{R} ^{8N \times 8N}$，which satisfies $f(R^\phi \cdot (x,y,t))=T^\phi\cdot f(x,y,t)$.
>
> Here $R^\phi$ denotes the rotation applied in longitude. The same reasoning applies to $\theta$ and time $\tau$, where additive shifts similarly induce orthogonal transformations within their respective subspaces. Therefore, spatial rotations and temporal shifts are exactly mirrored by orthogonal rotations in the embedding space, confirming that the 3D spherical RoPE is rotation-equivariant.
>
> > More comparative methods should be included. Although there may be no direct baselines for the entire task, fair comparisons can still be made at the "world initialization" and "world continuation" stages with recent panoramic video generation and interactive world synthesis methods (e.g., 4K4DGen, OmniDrag, WonderWorld).
>
> Thank you for pointing this out. We agree that comparisons at the world initialization and world continuation stages with recent relevant methods are valuable, and we appreciate the suggestions. Unfortunately, OmniDrag has not released its code, so we were unable to include a direct comparison.
>
> | Models  | SC-View | SC-All | MS-View | MS-All | AQ-View | AQ-All |
> | ------- | ------- | ------ | ------- | ------ | ------- | ------ |
> | 360I2V  | 92.5    | 94.8   | 98.9    | 98.7   | 49.5    | 55.0   |
> | 4K4DGen | 94.1    | 95.1   | 99.2    | 98.8   | 46.0    | 53.4   |
> | IaaW-CM | **95.8**    | **97.2**   | **99.3**    | **99.2**   | **49.7**    | **55.7**   |
>
> For 4K4DGen, which was proposed for animating 4K panoramic images, its code was not publicly available prior to our submission deadline, which prevented us from including a comparison at that time. Now that it is accessible, we have conducted an additional comparison with our IaaW-CM as below, focusing specifically on the world continuation stage. A direct comparison in the world initialization stage was not feasible due to differences in input format and task formulation. The results show that our model outperforms 4K4DGen in terms of both temporal consistency and visual quality. We attribute this to two factors:
> 1. Our model is trained on a large corpus of native equirectangular videos, allowing it to model the panoramic space holistically, including its inherent geometric distortions. This enables better spatial coherence across views.
> 2. In contrast, 4K4DGen divides the panoramic image into multiple blocks and animates them independently using a relatively weak image-to-video model (SVD), without any textual control. This process often results in spatial artifacts, temporal discontinuity, and semantic misalignment across blocks.
>
> We also thank the reviewer for mentioning WonderWorld, which we had not considered previously. WonderWorld focuses on static 3D scene generation using Gaussian-Splatting-like representation from a single image. While it does not support dynamic video generation as our method does, we will include a qualitative comparison in the revised version with our IaaW-IM in world initialization stage. Notably, WonderWorld does not produce equirectangular video but instead reconstructs a pseudo-3D splatting-based scene, which may enable longer exploration paths but often suffers from artifacts inherent to splatting (e.g., blur and point cloud sparsity). In contrast, our method generates temporally consistent and spatially coherent dynamic video with a spherical field of view, offering a more immersive and artifact-free experience.
>
> > Efficiency analysis. As an interactive world generation model, the reliance on VLMs and video diffusion raises concerns about inference and interaction latency. It is recommended to include experimental results assessing the model's efficiency.
>
> Thank you for this valuable comment, and we fully agree that efficiency is a key factor for practical deployment of any candidate pipeline for interactive visual world modeling. We report the efficiency of our pipeline as follows.
>
> | Model                     | Resolution   | Inference Time       | Inference Memory Usage    |
> | ------------------------- | ------------ | -------------------- | ------------------------- |
> | CogVideoX1.5-5B-I2V(base) | 1360\*768\*81  | ~16 min              | $\ge $9GB               |
> | 4k4DGen\*                  | 2048\*1024\*14 | ~16 min              | $\ge $12GB              |
> | IaaW-IM                   | 2048\*1024\*49 | ~(17*n) min n=\#views | $\ge $(12*n)GB n=\#views |
> | IaaW-CM                   | 2048\*1024\*49 | ~17 min              | $\ge $12GB              |
>
> $\small{\text{4k4DGen with \* here represents it can generate 4096*2048 resolution video, while for comparison, we include the 2048\*1024 results here.}}$
>
> Our method achieves inference times comparable to both the base foundation model (despite operating at higher resolution) and to prior work such as 4K4DGen (while generating more frames). Currently, each generation step (covering ~3–5 seconds of video) takes approximately 10–20 minutes on a single A100 GPU. This latency stems from the high output resolution (2048×1024) and the large size of the backbone models. As with most generative systems, there exists a trade-off between generation quality (e.g., resolution) and latency. In this work, we prioritize generation quality, though we also discuss various optimization techniques that could accelerate inference in our response to Reviewer 4SKk.

---

> > ### Comment · Reviewer_fetS · 2025-08-08
> >
> > Thank you for your detailed rebuttal. I appreciate the clarifications provided and have no further questions at this time.

---

> > > ### Author Response · Authors · 2025-08-08
> > >
> > > We appreciate your time and insightful review comments, which are very helpful for improving our paper.

---

### Official Review · Reviewer_GNKG · 2025-07-02

**Clarity:** 3
**Significance:** 3
**Originality:** 4
**Rating:** 5
**Confidence:** 3

**Summary:**

This paper introduces "Image as a World" (IaaW), a novel framework for generating interactive, 360-degree panoramic videos from a single input image. The core contribution is a three-stage pipeline designed to create a temporally and spatially coherent "world" that users can navigate. The process begins with World Initialization, which synthesizes an initial panoramic video from one image. This is followed by World Exploration, allowing user-controlled viewpoint rotation within the generated scene. Finally, World Continuation autoregressively extends the video sequence to create an endlessly explorable environment. To achieve this, the authors propose a diffusion-based visual world model enhanced with a novel 3D Spherical Rotary Positional Encoding (RoPE) to handle equirectangular geometry and a multi-view composition mechanism to improve scene diversity from the limited initial input. Additionally, a fine-tuned vision-language model, IaaW-VLM, generates dynamic, view-specific prompts to guide the generation process. The paper presents qualitative and quantitative results demonstrating the framework's ability to produce high-fidelity, controllable, and temporally consistent panoramic videos, outperforming baseline methods.

**Questions:**

1. About the controllability: Can a user control the content of the continued world by their own language commands (e.g., "move towards the mountains") instead of VLM auto-generated view prompts?

2. About long-term consistency: what is the maximum length of the generated video in practice?

**Ethical Concerns:**

["NO or VERY MINOR ethics concerns only"]

**Final Justification:**

After reading the rebuttal, my concerns have been resolved. I still find this work interesting and promising, and I am therefore inclined to recommend acceptance.

**Quality:**

3

**Strengths And Weaknesses:**

Strengths:

1. This paper is very well-written, clearly organized, and easy to follow.

2. The paper tackles a novel and ambitious problem: generating a complete, interactive, and infinitely extendable 360-degree world from a single image. This is a significant step beyond existing work in panoramic video generation, which often requires more comprehensive inputs (like full panoramic images or multi-view video) or lacks temporal continuity and interactive control. The problem formulation itself is a valuable contribution to the field of generative AI and world models.

3. The proposed IaaW framework is well-structured and technically sound. The three-stage pipeline (Initialization, Exploration, Continuation) is a logical approach to this complex problem.  The technical innovations are well-motivated:

4. The 3D Spherical RoPE is a clever adaptation of rotary embeddings for the specific geometry of panoramic videos, and the ablation studies show it effectively reduces spatial distortions.

5. The authors conduct a comprehensive evaluation with both qualitative and quantitative analyses.

Weaknesses:

1. While the presented examples (village streets, landscapes) are impressive, the paper could benefit from demonstrating the model's performance on a wider and more diverse range of scenes. How does it handle complex indoor environments or scenes with many moving objects?

2. The "World Exploration" stage allows for user-controlled viewpoint changes. However, during the "World Continuation" stage, it seems the progression of the scene is guided primarily by the IaaW-VLM's automatically generated prompts rather than explicit user commands.

---

> ### Author Rebuttal · Authors · 2025-07-31
>
> We sincerely appreciate the reviewer’s recognition of our contributions, particularly the technical innovations in both the overall pipeline and the proposed 3D Spherical RoPE. Below, we address the reviewer’s concerns and questions in detail.
> > While the presented examples (village streets, landscapes) are impressive, the paper could benefit from demonstrating the model's performance on a wider and more diverse range of scenes. How does it handle complex indoor environments or scenes with many moving objects?
>
> Thank you for your insightful comment. We agree that demonstrating the model's performance across a broader and more diverse set of scenes would strengthen the paper. Although we are unable to provide additional visualizations during the rebuttal phase according to the policy, we will include more results on diverse scenarios—including indoor and complex scenes—in the final version of the paper and supplemental material.
>
> Regarding performance on indoor environments, our model performs reasonably well. This is partly because a substantial portion of our training data includes game-rendered equirectangular videos, many of which are indoor scenes with consistent structure and lighting.
>
> However, performance on complex scenes with many moving objects is more limited. For example, in a case involving an aerial view of urban streets with a large number of cars, the generation results are not satisfactory: some cars remain static while others move in unnatural or incorrect directions. There are two main contributing factors:
>
> 1. Model Capacity: The base models we built on (including CogVideoX and other comparable open-source video diffusion models) struggle to robustly handle scenarios with multiple independently moving objects, which exceeds the temporal modeling capacity of existing models.
> 2. Training Data Bias: To ensure visual stability, we filtered out videos with significant camera shake—many of which were hand-held recordings containing dense motion and multiple objects. As a result, the training set is less representative of such complex dynamic environments, which impacts generalization.
>
> We acknowledge this limitation and plan to enhance both the model capacity and data diversity in future work to better handle these challenging scenarios.
>
> > The "World Exploration" stage allows for user-controlled viewpoint changes. However, during the "World Continuation" stage, it seems the progression of the scene is guided primarily by the IaaW-VLM's automatically generated prompts rather than explicit user commands.
>
> We appreciate the reviewer’s insightful comment. While the World Continuation stage does leverage the IaaW-VLM to automatically generate prompts for scene continuation, it is important to clarify that this stage can indeed be guided by explicit user instructions. The VLM is introduced to automate the repeatedly continuation process, but user-provided prompts can always override or guide the generation pipeline when desired. However, we acknowledge that the current model exhibits relatively limited instruction-following ability, especially when prompted with very short commands. This limitation arises because the training captions were generated using an automatic captioning model and were not specifically curated or aligned with diverse forms of user intent. We thank the reviewer for pointing out this issue. In future work, we plan to enhance the model’s instruction-following capabilities by incorporating more instruction-following captions, as well as refining the prompt processing method to better support flexible user control.
>
> > About the controllability: Can a user control the content of the continued world by their own language commands (e.g., "move towards the mountains") instead of VLM auto-generated view prompts?
>
> Thank you for this insightful question. We have conducted several preliminary trials with user-specified instructions such as "move towards the mountains." While our model can follow such commands to some extent, the behavior is not entirely reliable or consistent. This limitation primarily stems from the nature of our training data: the captions used for training were automatically generated and not explicitly curated to reflect diverse or instruction-driven user intents. As a result, the model lacks strong grounding in flexible, user-controlled language commands. In future work, we plan to enhance the model’s instruction-following capabilities by incorporating more instruction-following captions, as well as refining the prompt processing method to better support flexible user control.
>
> > About long-term consistency: what is the maximum length of the generated video in practice?
>
> Thank you for your interest in this aspect of our work. In practice, we have observed that performing world exploration and world continuation for around 5 to 10 rounds repeatedly can maintain reasonable quality. Specifically, with 10 rounds, we are able to generate a video approximately one minute long. However, when we extend the video beyond 10 rounds, the quality tends to degrade significantly. This degradation is primarily due to cumulative errors that arise during long-term generation, which is a known challenge in the field. The main limitation preventing longer extensions is that, in our current setup, generation relies solely on the most recent chunk due to memory constraints. To enhance long-term consistency, we are exploring techniques to better leverage historical frames—such as compressing them and incorporating as much of the historical context as possible during generation. We will include a discussion of this phenomenon and potential solutions in the revised version of the paper.

---

### Author Response · Authors · 2025-08-04

We sincerely appreciate all reviewers for your time and thoughtful feedback. We have carefully addressed each review comment in our rebuttal, and we remain available to provide additional clarification if you have any further questions during the discussion.

---

### Note · Authors · 2025-08-14

We thank the reviewers and Area Chair for their time and constructive feedback. We are encouraged that all reviewers acknowledged the novelty and value of our work on generating interactive, controllable 4D worlds from a single image.

Reviewers highlighted our main technical contributions, including the three-stage framework described as “well-structured and technically sound” (GNKG), and the 3D Spherical RoPE for ensuring geometric and temporal coherence, described as “innovative and relevant” (fetS, 8ork, 4SKk).

In our rebuttal, we added a theoretical proof for RoPE’s rotation equivariance (fetS), included additional comparisons with newly available 4K4DGen (fetS), and provided an efficiency analysis (fetS, 4SKk). We also clarified the model architecture, the role of the VLM, and how user control is implemented (GNKG, 8ork). In addition, we will extend the paper with a failure case analysis and a discussion of current limitations, such as long-term consistency and handling complex scenes (GNKG, 4SKk). We will incorporate these changes in the revised version.

We carefully addressed all concerns and no further questions are raised. We remain committed to improving the paper based on all feedback. We believe the planned revisions, mostly straightforward additions for clarity and comparison, will further strengthen the contribution. Thank you for your consideration.

---

### Decision · Program_Chairs · 2025-09-17

**Decision:**

Accept (poster)

**Comment:**

This paper presents a three-stage framework for turning a single image into an interactive, temporally extendable 360° world, with a well-motivated 3D Spherical RoPE, clear system design and demo views. GNKG and fetS are positive; 8ork raised clarity issues but increased the score to accept after the rebuttal. Reviewer 4SKk raised a few additional concerns after the rebuttal, but these were not posed to the authors for further reply. Upon careful review, the AC finds these points to be minor, and in several cases contradictory to other reviewers’ assessments. Therefore, AC recommend acceptance.